# PeerJ

# Morphometric comparisons of *Diaphorina citri* (Hemiptera: Liviidae) populations from Iran, USA and Pakistan

Mohammadreza Lashkari[1], Matthew G. Hentz[2] and Laura M. Boykin[3]

[1] Department of Biodiversity, Institute of Science and High Technology and Environmental Sciences, Graduate University of Advanced Technology, Kerman, Iran
[2] Port St. Lucie, Florida, United States of America
[3] The University of Western Australia, Australian Research Council Centre of Excellence in Plant Energy Biology and School of Chemistry and Biochemistry, Crawley, Perth, Western Australia, Australia

## ABSTRACT

The Asian citrus psyllid (ACP), *Diaphorina citri* Kuwayama (Hemiptera: Liviidae), vector of citrus greening disease pathogen, Huanglongbing (HLB), is considered the most serious pest of citrus in the world. Prior molecular based studies have hypothesized a link between the *D. citri* in Iran and the USA (Florida). The purpose of this study was to collect morphometric data from *D. citri* populations from Iran (mtCOI haplotype-1), Florida (mtCOI haplotype-1), and Pakistan (mtCOI haplotype-6), to determine whether different mtCOI haplotypes have a relationship to a specific morphometric variation. 240 samples from 6 ACP populations (Iran—Jiroft, Chabahar; Florida—Ft. Pierce, Palm Beach Gardens, Port St. Lucie; and Pakistan—Punjab) were collected for comparison. Measurements of 20 morphological characters were selected, measured and analysed using ANOVA and MANOVA. The results indicate differences among the 6 ACP populations (Wilks' lambda $= 0.0376$, $F = 7.29$, $P < 0.0001$). The body length (BL), circumanal ring length (CL), antenna length (AL), forewing length (WL) and Rs vein length of forewing (RL) were the most important characters separating the populations. The cluster analysis showed that the Iran and Florida populations are distinct from each other but separate from the Pakistan population. Thus, three subgroups can be morphologically discriminated within *D. citri* species in this study, (1) Iran, (2) USA (Florida) and (3) Pakistan population. Morphometric comparisons provided further resolution to the mtCOI haplotypes and distinguished the Florida and Iranian populations.

## INTRODUCTION

The Asian citrus psyllid (ACP), *Diaphorina citri* Kuwayama (Hemiptera: Liviidae) (*Burckhardt & Ouvrard, 2012*), is the vector of the bacteria 'Candidatus Liberibacter spp.,' the causal agent associated with Huanglongbing (HLB) or citrus greening disease (*Bové, 2006*; *Grafton-Cardwell, Stelinski & Stansly, 2013*; *Hall et al., 2013*). Huanglongbing is considered the world's most important disease of citrus (*Gottwald, 2010*; *Grafton-Cardwell,*

Corresponding author
Laura M. Boykin,
lboykin@mac.com

*Stelinski & Stansly, 2013*; *Hall et al., 2013*). The Asian citrus psyllid has been reported from the Arabian Peninsula, Afghanistan through to the Indian subcontinent, Japan, Taiwan, Hong Kong, China, the Philippine Islands, the Pacific Islands of Hawaii and Guam, the continental USA, the Caribbean, Central and South America and the Indian Ocean islands of Mauritius and Réunion (*Boykin et al., 2012b*). In southern Iran, the ACP was discovered in 1997 followed by the HLB disease in 2006 (*Bové et al., 2000*; *Faghihi et al., 2009*) and now it has established in the citrus growing regions of Hormozgan, Sistan–Baluchistan, Kerman and Fars Provinces. In the USA, it was first reported from Florida in 1998 (*Bové, 2006*) and now occurs from Florida to California (*Boykin et al., 2012b*). Also it was reported from Pakistan in 1927 (*Husain & Nath, 1927*), and has become a serious pest in all citrus growing areas of Pakistan (*Mahmood, Rehman & Ahmad, 2014*).

Worldwide genetic diversity of *D. citri*, based on mitochondrial cytochrome oxidase I (mtCOI) DNA sequences, suggests the existence of eight haplotypes (Dcit-1 to Dcit-8) (*Boykin et al., 2012b*). Haplotype-1 occurs in the following countries: United States of America (USA: Florida and Texas), India, Saudi Arabia, Brazil and Mexico. Haplotype 2 includes populations from Brazil (Sao Paulo), China (Fuzhou, Gangzhou), Indonesia (Java, Bali), Mauritius, Reunion, Taiwan (Taipei), Thailand (Hat Yai), and Vietnam (Hanoi). Haplotype 3 includes populations from Puerto Rico (Univ. of PR) and Guadeloupe. Haplotypes 4-8 includes populations from China (Gangzhou, Zhejaing), Florida and Mexico (Akil, Yucatan). An additional study revealed that *D. citri* populations from Iran are genetically similar to the mtCOI Haplotype-1 group, while the Pakistan population has been designated as mtCOI haplotype-6 (*Lashkari et al., 2014*). Further evidence supporting the haplotype grouping comes from *Wolbachia, wDi,* wsp sequences which indicated that the Iran population was similar to the Florida population, but was different from the Pakistan population (*Lashkari et al., 2014*).

Morphologically, the psyllids within *Diaphorina* can be differentiated by the shape of the genal processes the shape and pattern coloration of the forewings, the arrangement of spinules on the forewing membrane, and the shape of the female terminalia (*Hollis, 1987*). Six morphological measurements, including body length, wing length and width, genal process length and width, and antenna length have been used to study the morphometry of ACP populations on six Rutaceae from Mexico (*García-Pérez et al., 2013*). Additionally, *Vargas-Madríz et al. (2013)* used 4 morphological indices including body length, body width, wing length, and wing width to describe the morphometry of another psyllid species, *Bactericera cockerelli* (Hemiptera: Triozidae), on two varieties of host plant.

The purpose of this study was to explore whether the different mtCOI haplotypes of ACP populations (mtCOI haplotypes 1 and 6) have correlate with specific morphometric variation.

**Table 1** Collection sites, mtCOI Haplotype, hosts and number of examined specimens for populations of *Diaphorina citri*.

| Country | Province/State | County | mtCOI Haplotype | Host | n |
|---|---|---|---|---|---|
| Iran | Sistan & Baluchestan | Chabahar | 1 | *Citrus sinensis* (L.) Osbeck | 40 |
| | Kerman | Jiroft | 1 | *Citrus sinensis* (L.) Osbeck | 40 |
| USA | Florida | Palm Beach Gardens, Palm Beach County | 1 | *Murraya paniculata* (L.) Jack. | 40 |
| | | Port St. Lucie, St. Lucie County | 1 | *Murraya paniculata* (L.) Jack. | 40 |
| | | USDA ARS colony, Ft. Pierce, St. Lucie County | 1 | *Citrus macrophylla* Wester | 40 |
| Pakistan | Punjab | Punjab | 6 | *Citrus sinensis* (L.) Osbeck | 40 |

## MATERIAL AND METHODS

### Psyllid samples

Six genetic based populations of *D. citri* were collected from Iran (2) and Florida (3) as mitochondrial COI Haplotype 1, and Pakistan as mitochondrial COI Haplotype 6. The collected specimens were preserved in 96% ethanol (Table 1). The female adults were selected for this study for direct comparison to the previous molecular study (*Lashkari et al., 2014*), which included only females. Also, it has been shown that the structure of the male genitalia within *Diaphorina* is homogeneous throughout and species are defined on the shape of genal cones, the shape and coloration of the forewings, the arrangement of spinules on the forewing membrane, and the shape of the female terminalia (*Hollis, 1987*).

### Morphometric analysis

A total of 240 female adults (40 adults from each population) were randomly selected for morphological analyses. In order to calculate the morphometric information of the specimens, each insect was dissected to separate the different structures. The selected specimens were placed individually in 1.5 ml tubes containing 96% ethanol. Twenty standard morphological characters (Table 2) were selected to survey the morphometric variation among the six populations (*Burckhardt, 1986*; *Hollis, 1987*; *Ossiannilsson, 1992*; *Olivares & Burckhardt, 1997*; *Burckhardt & Basset, 2000*; *Mifsud & Burckhardt, 2002*; *García-Pérez et al., 2013*). Descriptions of the characters are given in Table 2 and Fig. 1. The body structures (except wings) were mounted on slides with glycerin and photographs were taken of each structure/specimen using a digital camera coupled with a stereomicroscope with 40X magnification. The right forewing of each specimen was slide-mounted using Euparal as the mounting medium. All measurements (mm) were performed with National Instruments Vision Assistant Software, version 2012 (*National Instruments Corporation, 2012*).

### Statistical methods

Data were analyzed using analysis of variance (ANOVA) to compare different populations for each character, and pairwise comparisons based on Tukey's HSD (Honest Significant Difference) test were calculated only after a significant ANOVA was found. A multivariate analysis of variance (MANOVA) was done for the comparison of the group means of

**Table 2 Morphological traits used for morphometric analysis of populations of *Diaphorina citri*.**

| Character no. | Acronym | Character |
|---|---|---|
| 1 | BL | Body length (from the apex of the genal process to the distal part of the proctiger) |
| 2 | HW | Head width |
| 3 | VW | Vertex width |
| 4 | VL | Vertex length |
| 5 | GL | Genal process length |
| 6 | GW | Genal process width |
| 7 | AL | Antenna length |
| 8 | WL | Forewing length |
| 9 | WW | Forewing width |
| 10 | RL | Rs vein length of forewing |
| 11 | RC | Length of the line connecting apices of vein Rs and Cu1a of forewing |
| 12 | a | Length of the line connecting the base and apex of vein M1 + 2 of forewing |
| 13 | b | Length of the line connecting the base and apex of vein M3 + 4 of forewing |
| 14 | c | Length of the line connecting apices of veins M1 + 2 and M3 + 4 of forewing |
| 15 | d | Length of the line connecting apices of vein Cu1a and Cu1b of forewing |
| 16 | e | Length of widest perpendicular distance to d in cell cu1 |
| 17 | ML | Metatibial length |
| 18 | FP | Female proctiger length |
| 19 | CL | Circumanal ring length |
| 20 | SL | Female subgenital plate length |

all variables. The Wilks' lambda test was applied as the statistical significance of the MANOVA. Moreover, the Canonical Variate Analysis (CVA) was used to determine the relative importance of characteristics as discriminators between groups. Mahalanobis distances (D2) were calculated between all populations' centroids using a pooled variance covariance matrix. All statistical analyses were conducted using the SAS statistical program (*SAS Institute, 2003*). The UPGMA (Unweighted Pair Group Method with Arithmetic Mean) hierarchical cluster analysis (*Sneath & Sokal, 1973*) based on squared Euclidean distances and the mantel tests were performed with NTSYS-pc program (*Rohlf, 1993*). Geographic distances among locations were measured using Google Earth (https://www.google.com/earth/).

## RESULTS

According to univariate analysis, 15 morphological characters were found to be significantly different among the six ACP populations ($\alpha = 0.01$). These included body length (BL) ($F = 78.07$, df = 5, $P < 0.0001$), Vertex width (VW) ($F = 4$, df = 5, $P = 0.0019$), antenna length (AL) ($F = 11.63$, df = 5, $P < 0.0001$), forewing length (WL) ($F = 28.50$, df = 5, $P < 0.0001$) and width (WW) ($F = 11.19$, df = 5, $P < 0.0001$), Rs vein length of forewing (RL) ($F = 26.92$, df = 5, $P < 0.0001$), length of the line connecting apices of vein Rs and Cu1a of forewing (RC) ($F = 7.27$, df = 5, $P < 0.0001$), length of the line connecting the base and apex of vein M3 + 4 of forewing (b) ($F = 6.38$, df = 5, $P < 0.0001$), length of

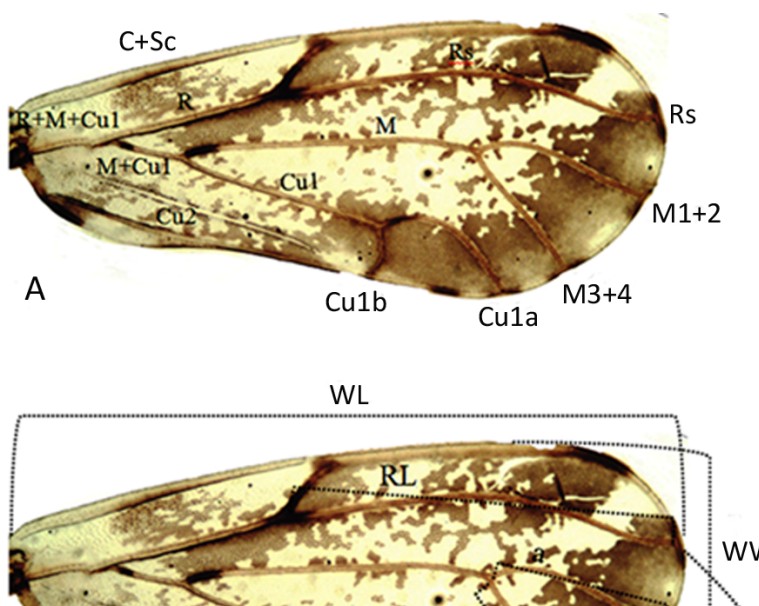

**Figure 1 Forewing vein terminology based on *Hodkinson & White (1979)*, and (B) lines indicating measurements based on *Mifsud & Burckhardt (2002)* in the right forewing of *Diaphorina citri*.** See Table 2 for abbreviations.

the line connecting apices of veins M1 + 2 and M3 + 4 of forewing (c) ($F = 9.69$, df $= 5$, $P < 0.0001$), length of the line connecting apices of vein Cu1a and Cu1b of forewing (d) ($F = 15.58$, df $= 5$, $P < 0.0001$), length of widest perpendicular distance to d in cell cu1 (e) ($F = 3.85$, df $= 5$, $P = 0.0025$), metatibial length (ML) ($F = 5.19$, df $= 5$, $P = 0.0002$), female proctiger length (FP) ($F = 14.69$, df $= 5$, $P < 0.0001$), circumanal ring length (CL) ($F = 15.74$, df $= 5$, $P < 0.0001$) and female subgenital plate length (SL) ($F = 8.94$, df $= 5$, $P < 0.0001$) (Table 3).

There was no statistical difference between populations with regards to the following characters: head width (HW), vertex length (VL), genal process length (GL), genal process width (GW) and length of the line connecting the base and apex of vein M1 + 2 of forewing (a).

The MANOVAs of the ACP populations revealed a significant difference among the size variables of the populations (Wilks' lambda $= 0.0376$, $F = 7.29$, $P < 0.0001$). The shortest Mahalanobis distance (D2 $= 0.720$) was between the two populations from Florida (Palm Beach Gardens and Port St. Lucie), whereas the longest distance was between the populations from Pakistan and Florida (Palm Beach Gardens) (D2 $= 36.756$) (Table 4). The cluster analysis revealed two major clusters. The first contained samples from Iran

**Table 3** Size (MM ± SE) of the 20 morphological traits in the populations of Asian citrus psyllid, *Diaphorina citri* from Iran, Florida and Pakistan.

| Variable[a] | Population | | | | | |
|---|---|---|---|---|---|---|
| | Iran—Chabahar | Iran—Jiroft | Florida—USDA ARS colony | Florida—Palm Beach County | Florida—St. Lucie County | Pakistan |
| BL | 2.477 ± 0.029 a[b] | 2.459 ± 0.029 a | 2.49 ± 0.018 a | 2.557 ± 0.009 a | 2.535 ± 0.013 a | 1.980 ± 0.035 b |
| HW | 0.575 ± 0.002 a | 0.569 ± 0.002 a | 0.558 ± 0.005 a | 0.573 ± 0.003 a | 0.567 ± 0.004 a | 0.573 ± 0.006 a |
| VW | 0.375 ± 0.003 a | 0.372 ± 0.003 ab | 0.359 ± 0.003 b | 0.376 ± 0.003 a | 0.370 ± 0.003 ab | 0.364 ± 0.004 ab |
| VL | 0.136 ± 0.002 a | 0.132 ± 0.002 a | 0.136 ± 0.003 a | 0.141 ± 0.002 a | 0.139 ± 0.003 a | 0.137 ± 0.004 a |
| GL | 0.122 ± 0.003 a | 0.121 ± 0.003 a | 0.124 ± 0.003 a | 0.132 ± 0.002 a | 0.128 ± 0.003 a | 0.122 ± 0.004 a |
| GW | 0.101 ± 0.002 a | 0.095 ± 0.002 a | 0.097 ± 0.002 a | 0.103 ± 0.002 a | 0.102 ± 0.002 a | 0.094 ± 0.003 a |
| AL | 0.447 ± 0.006 a | 0.441 ± 0.006 a | 0.436 ± 0.005 a | 0.447 ± 0.003 a | 0.442 ± 0.004 a | 0.403 ± 0.003 b |
| WL | 1.990 ± 0.014 c | 2.091 ± 0.012 b | 2.151 ± 0.011 ab | 2.172 ± 0.009 a | 2.164 ± 0.010 a | 2.090 ± 0.017 b |
| WW | 0.869 ± 0.007 b | 0.910 ± 0.006 a | 0.909 ± 0.004 a | 0.923 ± 0.003 a | 0.914 ± 0.004 a | 0.912 ± 0.006 a |
| RL | 1.135 ± 0.010 c | 1.222 ± 0.007 ab | 1.246 ± 0.008 ab | 1.261 ± 0.006 a | 1.250 ± 0.007 a | 1.206 ± 0.012 b |
| RC | 0.728 ± 0.008 b | 0.760 ± 0.007 a | 0.762 ± 0.005 a | 0.775 ± 0.005 a | 0.770 ± 0.005 a | 0.775 ± 0.007 a |
| a | 0.555 ± 0.005 a | 0.592 ± 0.005 a | 0.579 ± 0.004 a | 0.768 ± 0.179 a | 0.587 ± 0.003 a | 0.571 ± 0.006 a |
| b | 0.490 ± 0.005 b | 0.520 ± 0.004 a | 0.499 ± 0.003 b | 0.508 ± 0.001 ab | 0.504 ± 0.003 ab | 0.503 ± 0.005 ab |
| c | 0.319 ± 0.004 b | 0.336 ± 0.005 ab | 0.350 ± 0.004a | 0.355 ± 0.002 a | 0.353 ± 0.003 a | 0.337 ± 0.006 ab |
| d | 0.409 ± 0.004 c | 0.423 ± 0.005 cb | 0.440 ± 0.003 ab | 0.446 ± 0.002 a | 0.445 ± 0.003a | 0.426 ± 0.003 cb |
| e | 0.259 ± 0.002 b | 0.260 ± 0.003 ab | 0.259 ± 0.002 ab | 0.266 ± 0.002 ab | 0.264 ± 0.002 ab | 0.271 ± 0.003 a |
| ML | 0.563 ± .005 a | 0.557 ± 0.005 ab | 0.540 ± 0.003 b | 0.556 ± 0.003 ab | 0.548 ± 0.004 ab | 0.539 ± 0.004 b |
| FP | 0.511 ± 0.007 a | 0.509 ± 0.007 a | 0.479 ± 0.004 b | 0.495 ± 0.004 ab | 0.489 ± 0.004 ab | 0.450 ± 0.007 c |
| CL | 0.133 ± 0.001 ab | 0.130 ± 0.001 b | 0.133 ± 0.003 ab | 0.141 ± 0.002 a | 0.138 ± 0.003 ab | 0.116 ± 0.002 c |
| SL | 0.423 ± 0.005 a | 0.421 ± 0.005 a | 0.409 ± 0.004 a | 0.418 ± 0.003 a | 0.414 ± 0.003 a | 0.386 ± 0.005 b |

**Notes.**
[a] See Table 2 for abbreviations.
[b] Means with the same letter within each variable are statistically equal (Tukey, $P \leq 0.01$).

**Table 4** Mahalanobis distances among populations of *Diaphorina citri* from Iran, Florida and Pakistan.

| Population | Iran—Chabahar | Iran—Jiroft | Florida—USDA ARS colony | Florida—Palm Beach County | Florida—St. Lucie County | Pakistan |
|---|---|---|---|---|---|---|
| Iran—Chabahar | 0 | | | | | |
| Iran—Jiroft | 7.486 | 0 | | | | |
| Florida—USDA ARS colony | 18.277 | 9.264 | 0 | | | |
| Florida—Palm Beach County | 16.603 | 8.999 | 2.644 | 0 | | |
| Florida—St. Lucie County | 15.511 | 8.526 | 1.127 | 0.720 | 0 | |
| Pakistan | 24.478 | 23.119 | 32.071 | 36.756 | 33.397 | 0 |

(Jiroft, Chabahar) and Florida (Ft. Pierce, Port St. Lucie, Palm Beach Gardens) and the second one contained the Pakistan population (Fig. 2).

The canonical discriminant analysis indicated that the first two canonical variables (CVA1 and CV2) described 65.15% and 24.85% of the total variance, respectively. The first and second together (CVA1 + CVA2) equaled 90% (Table 5). The body length (BL),

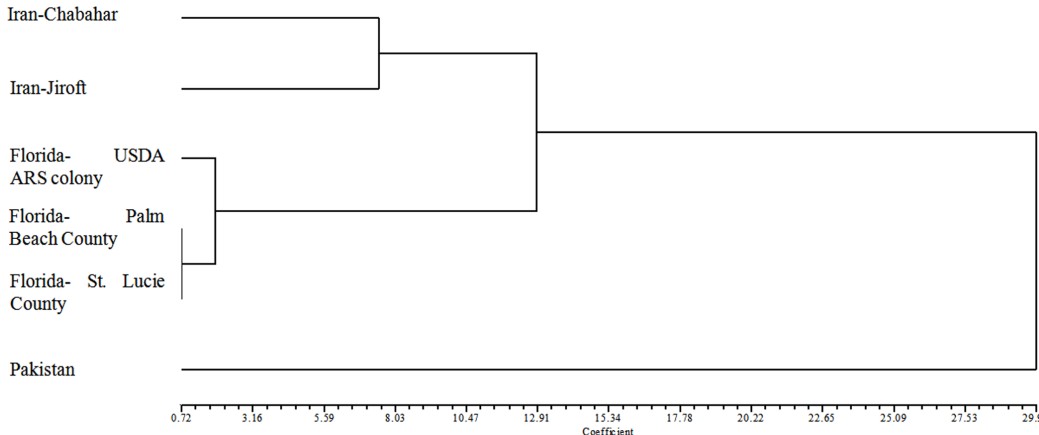

**Figure 2  Dendrogram plotted by UPGMA method based on squared Euclidean distance of *Diaphorina citri* populations.**

**Table 5  Standardized coefficients for canonical variables on the first (CVA1) and second (CVA2) canonical axes.**

| Variable[a] | CVA1 | CVA2 |
|---|---|---|
| BL | 0.853 | 0.390 |
| HW | −0.127 | 0.147 |
| VW | 0.068 | 0.261 |
| VL | 0.048 | −0.079 |
| GL | 0.151 | −0.107 |
| GW | 0.200 | 0.091 |
| AL | 0.453 | 0.356 |
| WL | 0.427 | −0.660 |
| WW | 0.151 | 0.498 |
| RL | 0.417 | −0.616 |
| RC | 0.039 | −0.462 |
| a | 0.093 | −0.046 |
| b | 0.057 | −0.094 |
| c | 0.317 | −0.446 |
| d | 0.390 | −0.518 |
| e | −0.156 | −0.263 |
| ML | 0.083 | 0.398 |
| FP | 0.307 | 0.572 |
| CL | 0.576 | 0.160 |
| SL | 0.340 | 0.408 |
| Eigenvalues | 4.4741 | 1.7065 |
| Proportion | 0.6515 | 0.2485 |

**Notes.**
[a] See Table 2 for abbreviations.

circumanal ring length (CL), antenna length (AL), forewing length (WL) and Rs vein length of forewing (RL) contributed most to this variation based on the first canonical axes (CVA1). Other characteristics also contributed, but to a lesser extent (Table 5).

The Mantel test showed that there was not significant correlation between geographic and morphological distances ($r = 0.535$, $p = 0.999$). Therefore, geographical distances did not impact the morphological differentiation found between the populations.

## DISCUSSION

The morphometric analyses of the ACP populations from Iran, USA (Florida) and Pakistan indicated the existence of two main groups within the populations analyzed. The first group included populations from Iran (Jiroft and Chabahar) and the USA (Florida), and the second was represented by a population from Pakistan. These results support similar findings from wing structures of ACP from Iran and Pakistan (*Lashkari et al., 2013*). The results presented here also support previous findings indicating that *D. citri* populations in Iran and Florida are similar and separated from Pakistan populations based on a global phylogenetic analyses of mtCOI, and *Wolbachia* wsp sequences (*Lashkari et al., 2014*). Prior molecular based studies showed that all Iranian populations of ACP are genetically similar to the Florida populations indicating a link between the ACP in Iran and the USA (Florida) (*Lashkari et al., 2014*).

The morphometric data provides further resolution to the previous molecular research, which indicated that different mtCOI haplotypes of ACP populations (mtCOI haplotypes 1 and 6) correlate with specific morphometric variation. The Iran and USA (Florida) populations (Haplotype 1) were distinguishable from Pakistan population (Haplotype 6) using mtCOI. Understanding the link between morphological and molecular characters is of vital importance for designing diagnostic tests for highly invasive species to aid global biosecurity (*Boykin et al., 2012a*; *Boykin et al., 2012b*).

The Mantel test results showed that the separation of the Iran and Florida populations in this study was not due to the geographic distance. *García-Pérez et al. (2013)* have shown the separation of host-associated populations of ACP. They showed that the host species or variety can influence morphometric traits of different host associated populations of ACP (*García-Pérez et al., 2013*). The largest ACP populations were associated with *C. sinensis* (L.) Osbeck cv. 'Marrs,' *C. sinensis* (L.) cv. 'Valencia' and *Murraya paniculata* (L.) Jack, while, the smallest sizes were found in males collected from *Citrus limetta* Risso, *C. sinensis* (L.) 'Selection 8' and *C. paradisi* Macfad. In the present study the populations from Iran that were collected from the same host species (*C. sinensis*) that were clustered together and the Florida populations that were collected from *M. paniculata* were clustered together, while the Florida population collected from *C. macrophylla* was separate. The main purpose of the present study is to explore whether the mtCOI haplotypes 1 and 6 of ACP populations have specific morphometric variation. We investigated the morphological characteristics of populations from Iran and Florida as Haplotype-1 *Diaphorina citri*, and the Pakistan population as *D. citri* Haplotype-6 that were defined in *Lashkari et al. (2014)*.

The current study does not incorporate potential intraspecific morphometric variation among different generations or host plants.

In the current study, the body length (BL), circumanal ring length (CL), antenna length (AL), forewing length (WL) and Rs vein length of forewing (RL) contributed most to the variation found among the six populations. These results were similar to a previous study conducted by *García-Pérez et al. (2013)*. They indicated that wing length, wing width and body length were the main variables contributing to discrimination of populations of *D. citri* on various host plants in Mexico. A comparison of the female ACP body size from the populations in the current study with those from different countries (Mexico, Réunion, Venezuela, and India) indicated that the populations from Iran and Florida were most similar to those from India (body length 2.4 mm; forewing length 2.17 mm), while the Pakistan populations stood alone while being shorter than the others (*García-Pérez et al., 2013*; *Étienne et al., 2001*; *Fonseca, Valera & Vasques, 2007*; *Mathur, 1975*; *Chhetry, Gupta & Tara, 2012*).

There are two pieces of evidence that suggest the invasion of ACP into Iran and the USA (Florida) originated from southwestern Asia, particularly India: 1- Southwestern Asia, i.e., India, has been suggested as the origin of ACP based on plant host origins and historical information (*Hall, 2008*); 2- The mitochondrial haplotype network for *D. citri* suggests a basal and thus ancestral position for Dcit-1 haplotype (*Boykin et al., 2012b*). *Boykin et al. (2012b)* showed that the Indian, USA, Saudi Arabian, Brazilian and Mexican populations of ACP belong to the mtCOI Haplotype-1 group. However, additional studies based on the morphological and other molecular markers such as microsatellite on the phylogenetic relationships among worldwide ACP populations are needed to confirm our hypothesis. *Boykin et al. (2007)* developed twelve polymorphic microsatellite markers for ACP and should be explored on a global scale.

The differentiation of populations may originate from one of the following events: insect migration, a new host or a new habitat or both of them, landscape changes (bottleneck effect), and genetic changes by stochastic events, such as gene flow, genetic drift and mutation or natural selection (*Kim & McPheron, 1993*; *Berlocher & Feder, 2002*). These variations may be changes in morphology, physiology, behavior, and life history traits, and subsequently would lead to the manifestation of the different taxonomic status of local populations such as biotype and ecotype (*Kim & McPheron, 1993*).

We conclude that *D. citri* populations related to the mtCOI haplotypes-1 (Iran and Florida) and 6 (Pakistan) have distinct morphometric characters based on multivariate analysis of morphological data. Future ACP studies are needed to confirm the relationship found here between the mtCOI haplotypes and morphology.

## ACKNOWLEDGEMENTS

We are also grateful to Dr. Shahid Nadeem Chohan (COMSATS Institute of Information Technology, Islamabad, Pakistan) for providing insect samples. LMB and MH would like to thank Mary Boykin for sample transport assistance.

### Funding

Financial support (No. 7.678) was provided by the Institute of Science and High Technology and Environmental Sciences, Graduate University of Advanced Technology, Kerman, Iran. The funders had no role in study design, data collection and analysis, decision to publish, or preparation of the manuscript.

### Grant Disclosures

The following grant information was disclosed by the authors:
Institute of Science and High Technology and Environmental Sciences: 7.678.
Graduate University of Advanced Technology.

### Competing Interests

Laura Boykin is an Academic Editor for PeerJ.

### Author Contributions

- Mohammadreza Lashkari conceived and designed the experiments, performed the experiments, analyzed the data, contributed reagents/materials/analysis tools, wrote the paper, prepared figures and/or tables, reviewed drafts of the paper.
- Matthew G. Hentz performed the experiments, contributed reagents/materials/analysis tools, wrote the paper, reviewed drafts of the paper.
- Laura M. Boykin conceived and designed the experiments, analyzed the data, contributed reagents/materials/analysis tools, wrote the paper, reviewed drafts of the paper.

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
