# Peer review of "Morphometric comparisons of Diaphorina citri (Hemiptera: Liviidae) populations from Iran, USA and Pakistan"

_PeerJ, doi:10.7717/peerj.946_

## Round 0.1 · original submission · Major Revisions

Please address the comments made by both reviewers, especially the major ones by Reviewer 2 regarding: (1) the number of populations selected for this study; (2) the potential influence of polyvoltinism on morphological variation (a point also hinted at by Reviewer 1); and (3) using measurements only from females. Reviewer 2 also made additional comments directly to the manuscript.

I also have a few major comments:

1. The number of populations shifts from 6 to 4 in the study. Please correct or clearly explain this shift in the text.

2. I do not think the data show support for the conclusion that " in most variables, the ACP populations from Iran and Florida ... differentiated from the Pakistan population" (lines 144-5). I count only 5 of the 20 characters in the Pakistan population (BL, AL, FP, CL, SL) that are statistically different from all measurements in the Iran/Florida populations.

3. The general discussion in lines 176-192 about conflicts in morphological versus molecular phylogenies does not seem relevant to the current paper. From what you have written, your morphometric phylogeny does not differ from the previous molecular result, but rather it is more resolved (yet not conflicting). So there is no need for this long discussion about how other studies show different molecular versus morphological phylogenies, because your study does not show this conflict.

Other minor comments:

4. Fig. 2 is not cited anywhere in the manuscript.

5. A few sentences are incomplete (line 83; 234) or do not make sense (lines 215-6).

6. line 225: Are these measurements mean values? I assume more than one individual was measured?

7. The reference to Google Earth (line 126) needs a url, access date and/or citation.

8. line 63: Replace “consists of” with “occurs in”

Reviewer 1 ·

Basic reporting

This manuscript is well down and describes a study investigating morphometric traits with genetic origins of populations. The study is well down and data analyzed appropriately. This study certainly deserves publication and will contribute to our knowledge. While insects were collected over a period of a year, it would be interesting to know general months of collection. While there may possibly be environmental effects on morphometric traits, it is important to evaluate the impact of genetic origin (as done in this study)

Experimental design

The experimental design is appropriate for the study.

Validity of the findings

The findings are valid and discussed appropriately. Statistical analysis was appropriate and correctly interpreted.

Additional comments

References . Several references list the issue of the journal and that should be removed (lines 239, 264, 268, 271, 280, 284, 295, 298, 302, 309 and 318). Italics are missing on a scientific name (line262), capitalize a word (Indian line 299), write out all of the journal name (line 273) and remove capitals on each word in title (line 315).
Line 193, add the in front of Pakistan, line 193, change is to was. The sentence on line 193-194 is not clear at all, rephrase. Line 219, missing ‘in’ in front of morphology

·

Basic reporting

This paper compares a few populations of Diaphorina citri using morphometric analyses. The number of populations compared is very small (5) making the result of limited validity. Choosing more populations may have produced a much more complex and probably different picture.

Experimental design

Diaphorina citri is polyvoltine. The authors should test if the different generations of D. citri over the year show morphometric differences. If there are morphometric seasonal differences within one population then the comparison of different populations becomes problematical.
It seem that the authors investigated only females. Why? Both sexes should be investigated separately and the results compared.

Validity of the findings

The authors acknowledge the importance of the host species/variety as influence of the size in D. citri but in the analysis specimens from different hosts and localities are mixed.

Additional comments

Additional comments in the attached manuscript.

---

## Round 0.2 · Minor Revisions

Only one remaining comment - In accordance with the reviewer, I suggest adding at least a sentence or two to the discussion, pointing out that the current study does not incorporate potential intraspecific morphometric variation among different generations or host plants (somewhere within the third paragraph of the discussion [lines 174-187] might be a good spot for this).

·

Basic reporting

The authors adressed most of the comments.

Experimental design

see general comments

Validity of the findings

see general comments

Additional comments

While most of the comments on the first manuscript version have been addressed by the authors, there are three points where I would like to comment: choice of females, variation between generations and influence of host races.
In all three cases the authors argued that their choice of material for the analyses was made for the best comparability with the molecular studies.
I appreciate this argument and it makes sense for the choice of females. The reason is that the mitochondrial DNA is passed along the female line. The morphological explanation by the authors is wrong as they are talking about interspecific and not intraspecific variation.
In the case of variation the morphological variation between different generations and hosts the argument is not valid as these are not genetically regulated. This means that the analysed morphometric data could have been also influenced by these factors. The results and conclusions of this paper are, therefore, very limited. The authors should clearly state these limitations.
Daniel Burckhardt

---

## Round 0.3 · accepted · Accept

Recommend to accept the manuscript for publication.